# Individual Characteristics as Prognostic Factors of the Evolution of Hospitalized COVID-19 Romanian Patients: A Comparative Observational Study between the First and Second Waves Based on Gaussian Graphical Models and Structural Equation Modeling

**DOI:** 10.3390/jcm10091958

**Published:** 2021-05-02

**Authors:** Alexandra Mocanu, Gratiela Georgiana Noja, Alin Viorel Istodor, Georgiana Moise, Marius Leretter, Laura-Cristina Rusu, Adina Maria Marza, Alexandru Ovidiu Mederle

**Affiliations:** 1Department XIII, Discipline of Infectious Diseases, “Victor Babes” University of Medicine and Pharmacy Timisoara, 2 Eftimie Murgu Square, 300041 Timisoara, Romania; alexandramocanu021@yahoo.com; 2Department of Marketing and International Economic Relations, Faculty of Economics and Business Administration, West University of Timisoara, 16 Pestalozzi Street, 300115 Timisoara, Romania; gratiela.noja@e-uvt.ro; 3First Department of Surgery, Second Discipline of Surgical Semiology, “Victor Babes” University of Medicine and Pharmacy Timisoara, 2 Eftimie Murgu Square, 300041 Timisoara, Romania; 4Department of Clinical Pharmacology, “Victor Babes” University of Medicine and Pharmacy, “Pius Brinzeu” County Emergency Clinical Hospital Timisoara, 2 Eftimie Murgu Square, 300041 Timisoara, Romania; drgeorgianamoise@gmail.com; 5Department of Prosthodontics, Multidisciplinary Center for Research, Evaluation, Diagnosis and Therapies in Oral Medicine, “Victor Babeș” University of Medicine and Pharmacy Timisoara, 2 Eftimie Murgu Square, 300041 Timisoara, Romania; 6Department of Oral Pathology, Multidisciplinary Center for Research, Evaluation, Diagnosis and Therapies in Oral Medicine, “Victor Babeș” University of Medicine and Pharmacy Timisoara, 2 Eftimie Murgu Square, 300041 Timisoara, Romania; laura.rusu@umft.ro; 7Department of Surgery, Multidisciplinary Center for Research, Evaluation, Diagnosis and Therapies in Oral Medicine, “Victor Babes” University of Medicine and Pharmacy Timisoara, 2 Eftimie Murgu Square, 300041 Timisoara, Romania; marza.adina@umft.ro (A.M.M.); mederle.ovidiu@umft.ro (A.O.M.)

**Keywords:** coronavirus pandemic, COVID-19, disease, treatment, strategy, econometric modeling

## Abstract

This study examines the role played by individual characteristics and specific treatment methods in the evolution of hospitalized patients with coronavirus disease 2019 (COVID-19), through the lens of an observational study performed in a comparative approach between the first and second waves of coronavirus pandemic in Romania. The research endeavor is configured on a two-fold approach, including a detailed observation of the evolution of 274 hospitalized patients with COVID-19 (145 in the first wave and 129 in the second wave of infection) according to specific treatment methods applied and patients’ individual features, as well as an econometric (quantitative) analysis through structural equation modeling and Gaussian graphical models designed to acknowledge the correlations and causal relationship between all considered coordinates. The main results highlight that the specific treatment methods applied had a positive influence on the evolution of COVID-19 patients, particularly in the second wave of coronavirus pandemic. In case of the first wave of COVID-19 infection, GGM results entail that there is a strong positive correlation between the evolution of the patients and the COVID-19 disease form, which is further positively correlated with the treatment scheme. The evolution of the patients is strongly and inversely correlated with the symptomatology and the ICU hospitalization. Moreover, the disease form is strongly and inversely correlated with oxygen saturation and the residence of patients (urban/rural). The symptomatology at first appearance also strongly depends on the age of the patients (positive correlation) and of the fact that the patient is a smoker or non-smoker and has other comorbidities. Age and gender are also important credentials that shape the disease degree and patient evolution in responding to treatment as well, our study attesting strong interconnections between these coordinates, the form of disease, symptomatology and overall evolution of the patients.

## 1. Introduction

The coronavirus disease 2019 (COVID-19) pandemic was first identified in late 2019 (November–December) in Wuhan, China and rapidly spread around the globe affecting millions of people worldwide, transforming into a public health emergency. The cause of this outbreak is a new virus, known as severe acute respiratory syndrome coronavirus 2 (SARS-CoV2). Due to its decisive importance, the research on this new type of coronavirus needs to be complemented and enhanced with comprehensive studies in order to strengthen the knowledge in this scientific field.

Coronaviruses are a group of RNA related viruses that cause the disease in birds and mammals. In humans, it causes respiratory infections that can range from mild infections (common cold) to severe forms (SARS, MERS, COVID-19) [1].

The SARS-CoV2 virus was detected by RT-PCR investigation (reverse transcriptase polymerase chain reaction). It has been isolated from nasal exudate, pharyngeal exudate, urine, feces, and lung tissue biopsy. It has been shown that SARS-Cov2 infection is not limited to the respiratory tract. The incubation period of the virus varies between 2 to 10 days, with an average of 6 days and a maximum of 12 days [2,3,4].

There are several biological markers used to identify the severity of the disease in patients with SARS-Cov2 infection. These include lymphopenia, d-dimers, ferritin, LDH, IL-6, CRP, fibrinogen, thrombocytopenia, procalcitonin [5].

Lymphocytopenia is a prominent feature of critically ill patients with SARS-Cov2 infection. This happens because of a targeted invasion by viral particles that create damage in the cytoplasm of the lymphocyte [6]. D-dimer is a fibrin degradation product, also known as FDP. A fragment from a small protein present in the blood after the blood clot is degraded by fibrinolysis shows the presence of this marker. Elevated D-dimer values are a marker of severe prognosis for COVID-19 infection, with an increased risk of mortality. The most common complications that can occur due to increased D-dimer values are pulmonary thromboembolism, disseminated intravascular coagulation (DIC), deep vein thrombosis and entero-mesenteric infarction (mesenteric ischemia) [7]. Ferritin has an important role as it limits the supply of iron. It is excessively produced in acute inflammatory diseases; therefore, high levels of ferritin are found in patients with pathogenic conditions. C-reactive protein (CRP) is a compound synthesized by the liver in acute inflammations. CRP and serum ferritin both play important roles in producing proinflammatory cytokines. LDH is an enzyme with a role in energy production, found in almost all the cells in the body. The concentration of LDH is associated with various tissues damage, including liver and lung. Increased levels are correlated with cell destruction and high levels of LDH found in patients with SARS-Cov2 infection show lung damage [8]. *Procalcitonin* (PCT) is the most useful marker of severe systemic inflammation, produced by the C-cells of the parathyroid gland. During infection, neuroendocrine cells from the lungs and intestine, mediated by proinflammatory cytokines (TNF alpha and IL-6), also produce PCT. Therefore, patients with viral infections show higher levels of PCT [9]. *Fibrinogen*, a glycoprotein produced by the liver, has an important role in regulating the inflammatory response. One of the complications of SARS-Cov2 infection is hypercoagulability, which includes high levels of fibrinogen. In an acute phase, fibrinogen contributes in decreasing the inflammation as it acts as a ligand for leucocyte integrin Mac-1, which is a surface receptor for extracellular double stranded RNA. As SARS-Cov2 is an RNA virus, fibrinogen competes with the virus in binding to these receptors. Additionally, at this stage, thrombi formation occurs due to increased levels of fibrinogen. *Cytokine storm and IL-6*—interferon-γ, interleukin-1, interleukin-6, TNF, and interleukin-18 are key cytokines that have high serum levels in the cytokine storm and are thought to have central immunopathological roles. Fever, a clinically distinctive sign of the cytokine storm, can be caused by interleukin-1, interleukin-6, or TNF by distinct mechanisms. *Pulmonary CT* scan is much more specific to highlight abnormalities in patients with SARS-Cov2 infection and describes the morphological features of opacities. Bilateral pulmonary opacities occur frequently having a distribution in the peripheral lower part of the lung. In typical cases, there are bilateral opacities of ground glass, sometimes with areas of consolidation and may have a suggestive appearance for the organization of pneumonia [10].

Acknowledging these facts and challenges, the main objective of the study is to explore the specific ways in which hospitalized COVID-19 patients are recovering depending on individual characteristics and personal/environmental features, disease form, associated comorbidities, and methods of treatment. We describe epidemiological, clinical, laboratory, and radiological characteristics, treatment, and outcomes of hospitalized COVID-19 patients. Furthermore, using a modern and complex methodology based on network analysis through Gaussian graphical models (GGMs) and structural equation modeling (SEM), we aim to bring new evidence on how the evolution of COVID-19 patients can be improved through comprehensive approaches tailored based according to individual features and treatment methods. Despite a better understanding of SARS-COV2 and despite treatment modification, we observed a much more aggressive virus in the second wave, with higher mortality, more severe cases as well as less asymptomatic cases.

This research strengthens the literature with robust evidence encompassed by a two-fold research endeavor centered on both a clinical/medical observation of COVID-19 patients and a complex econometric assessment of the interlinkages between patient evolution, personal traits (age, gender, residence, health status, comorbidities) and treatment methods applied.

## 2. Materials and Methods

### 2.1. Data—Sample, Features, Compilation

We retrospectively assessed the imaging and clinical characteristics and biological samples of a total of 274 hospitalized patients during the first two waves of infection with SARS-COV2 that appeared in Romania in 2020. The patients comprised in our samples were selected from the Infectious Diseases and Pneumo-Phthisiology Hospital Timisoara and further considered in the assessment after the confirmation of SARS-COV2 through the positive results of the exudate (RT-PCR COVID-19), being designated for the treatment of the aforementioned disease. In the time frame included in this study, many hospitals in Romania received symptomatic and asymptomatic patients. The asymptomatic patients did not present any symptoms and they were only admitted based on a positive PCR test. During those months, hospitals in Romania (including Infectious Diseases and Pneumo-Phthisiology Hospital Timisoara) served also as isolation units and this is the reason why asymptomatic patients were checked in.

In the first wave of infection with COVID-19, we analyzed 145 hospitalized patients with SARS-COV2 infection confirmed by RT-PCR, of which 76 men. Out of the 145 hospitalized patients considered, 6 died and 10 needed invasive mechanical ventilation, being treated in the intensive care unit—ICU department.

The second wave, much more aggressive, comes with certain changes in terms of therapy and patient behavior. We examined a group of 129 patients, of which 61 men. In total, 24 patients died and 13 needed intensive care in the ICU department.

Table 1 summarizes the above detailed data by presenting individual features, demographics and other characteristics of both samples and also enables to view the two waves side-by-side. The acronyms and description of all the variables/indicators used in this research are presented in the Appendix A, Table A1.

Descriptive statistics of the other variables captured in our empirical analysis in both samples are detailed in Table 2 and Table 3.

We collected and analyzed the following biological samples: leukocytes, lymphocytes, hematocrit, hemoglobin, platelets, fibrinogen, D-dimers, prothrombin time, INR, urea, creatinine, SGOT, GPT, sodium, potassium, total bilirubin, troponin, ferritin, glycemia, LDH, procalcitonin, lactate, CRP, IL-6, Clostridium difficile toxina A/B. From a medical imaging point of view, pulmonary CT was performed in patients from both groups. We primary analyzed the symptomatology of patients at admission into the hospital and at discharge, place of residence, the number of days of hospitalization, the number of days until the negative result of the exudate (RT-PCR COVID-19), distribution by sex, and new pathologies associated at discharge. Both groups received antiviral medication, corticotherapy, anticoagulant and antibiotic therapy as needed, according to the protocol in force at the time. The results obtained after analyzing these biological samples are presented in Section 3.

### 2.2. Methodology—Model, Testing and Validation

The econometric analysis performed in this paper is based on a modern methodological endeavor encompassing the network analysis through Gaussian graphical models (GGMs) and structural equation modeling (SEM) configured to identify and evidence the patterns and overall linkages between the credentials assessed in both samples of patients compiled for the first and second waves of COVID-19 in Romania during 2020.

The network analysis is designed through Gaussian graphical models (GGMs) which are innovative exploratory research tools, extremely useful to infer the connections between variables (including individual features and medical credentials). The fundamental advantage of the GGMs is represented by the ability to handle different types of variables/data, as comprised in our dataset, since the indicators included in the empirical analysis have different measurement units (e.g., continuous, binary, multi-category). For this research, GGMs were designed and processed through the extended Bayesian information criteria (EBIC) with graphical (g) least absolute shrinkage and selection operator (lasso) (EBICglasso) and partial correlation (Pcor). A Gaussian graphical model (GGM) is a “graph in which all random variables are continuous and jointly Gaussian [11,12] and it is based on conditional independence, respectively if $\Omega =\left\{\omega_{v1,v2}\right\}$, two variables v1 and v2 are conditionally independent if $\omega_{v1,v2}=0$, namely there are 0 entries of the precision matrix $\Omega =\Sigma^{-1}$. Σ is the positive definite covariance matrix and Ω is the precision matrix of the distribution, defined as the inverse of Σ”.

“If Σ is positive definite, distribution has density on $f(x|\xi ,\Sigma )={\left(2\pi \right)}^{-d/2}{\left(\det\Omega \right)}^{1/2}e^{-{\left(x-\xi \right)}^{T}\Omega (x-\xi )/2}$. The sample covariance matrix is represented by $\overset{-}{\Sigma }=\frac{1}{n-1}{\sum_{i=1}^{n}\left(x_{i}-\xi )(x_{i}-\xi \right)}^{T}$” [11,12,13].

GGMs allowed us to capture “conditional associations and avoid spurious correlation, grasping an undirected network of partial correlation coefficients (both positive visualized with blue edges and inverse captured with red edges), graphically reflected through the absolute strengths, width and saturation of the edges between nodes” [13].

Furthermore, GGMs are complemented in current research by another complex technique designed to model longitudinal data, namely structural equation modeling (SEM). SEM brings forward our empirical analysis since it re-joins path analysis, factor analysis and regression, and therefore it allows to specify and assess multiple causal associations between the constructs. Therefore, we were able to model conditional associations, namely the degree in which the variables are independent after conditioning on all other variables in the data set. This feature was essential in our empirical research endeavor since we focused on 17 major credentials that capture, in a gradual frame, main coordinates, patterns, evolution of the COVID-19 disease in the case of 274 hospitalized patients considered in our analysis. All of these credentials need to be assessed in their tight interdependence and sequential approach, as a complex network (performed in this paper through the GGMs) and through causal relationships (as enhanced by the SEM models designed to achieve the complicated model setup).

Structural equation modeling (SEM) represents a complex research methodology applied in this study to identify and evaluate direct and latent interlinkages between several specific variables associated with the COVID-19 disease and the evolution of the patient, so as to evidence the positive and negative influences in this regard. By the “evolution of the patient” (EP) we refer to deaths (unfavorable) or no deaths (favorable). Not all patients that developed serious complications and were brought in ICU department have died and for this reason EP is a separable variable than ICU. The general design of the SEM models configured in our research is presented in Figure 1.

As a complex multivariate analysis technique, often used in social sciences, structural equation modeling is applied in this medical research to evidence the relationship between observed variables in both samples of patients in a comparative approach between the first and second waves of infection. “Going beyond the classical linear regression analyses, SEM examines the causal relationships among variables, while controlling simultaneously for measurement error as a greatest advantage in empirical researches. SEM allowed us to determine the degree of correlation (path coefficients) that capture the importance of a certain path of influence from cause to effect” ([13], p. 6).

## 3. Results

### 3.1. Results of the Clinical/Medical Detailed Observation and Monitoring of COVID-19 Patients from the Two Waves of Infection

In the first wave of infection, 145 patients were analyzed, while in the second wave of SARS-COV2 infection we examined 129 patients. The patients comprised in both samples were selected from the Infectious Diseases and Pneumo-Phthisiology Hospital Timisoara and further considered in the assessment after the confirmation of SARS-COV2 through the positive results of the exudate (RT-PCR COVID-19), being designated for the treatment of the aforementioned disease.

Symptomatology at the onset of SARS-COV2 infection and during hospitalization are summarized in a comparative approach between the two waves of infection, as presented in Table 4.

Biological samples from 274 hospitalized patients and specific treatment methods applied are also presented in a comparative perspective in Table 5.

Patients in the first wave of COVID-19 infections were treated with: antivirals (25.65% of them with lopinavir + ritonavir), 24.58% with darunavir + cobicistat, 41.13% with darunavir + ritonavir, and 28.96% with hydroxychloroquine), antibiotic therapy (azithromycin—35.17%, vancomycin—2.78%), corticotherapy—31%, anticoagulant—21.37%. The second wave, much more aggressive, brought some changes in the therapeutic behavior of patients. In addition to the antivirals used in the first batch, lopinavir + ritonavir, darunavir + cobicistat, darunavir + ritonavir were used, for the medium to severe forms that amounted to 31% remdesivir was used and for the medium forms of the disease (12.50%) favipiravir was used as an antiviral, depending on their availability. These antivirals were used in combination with immunomodulators: Anakinra—4.65% and tocilizumab—15.50% and corticotherapy—77.51%.

### 3.2. Results of the Empirical Network Analysis—Gaussian Graphical Models (GGMs)

The network analysis was performed based on Gaussian graphical models (GGMs) configured on each dataset/sample of COVID-19 patients through two different methods targeting conditional dependence, namely the partial correlation (pcor) and extended Bayesian information criteria (EBIC) with least absolute shrinkage and selection operator (EBICglasso). The results of the first set of GGMs configured through partial correlations are entailed in Figure 2 and the results of the second set of GGMs estimated through EBICglasso are presented in Figure 3.

Main purpose of the network analysis through GGMs is to examine the presence, width and saturation of the connections between all considered credentials associated with our study and specific for the COVID-19 disease.

In case of the first wave of COVID-19 infection, GGM results entail that there is a strong positive correlation between the evolution of the patients (EP) and the COVID-19 disease form (DF) developed by the hospitalized patients, which is further positively correlated with the Rezolsta treatment scheme and inversely correlated with DRV + RTV treatment method. Further positive linkages are with age and gender of the patients (a positive connection but with a lower intensity). The evolution of the patients is strongly and inversely correlated with the symptomatology and the ICU hospitalization. Moreover, the disease form is strongly and inversely correlated with DRV + RTV treatment and further negatively correlated with oxygen saturation (SPO2) and the residence of patients (U/R, urban/rural). The symptomatology at first appearance also strongly depends on the age of the patients (positive correlation) and of the fact that the patient is a smoker or non-smoker (S/NS) and has other comorbidities (CMD).

In the second wave of COVID-19 infection, these interlinkages seem to be less intense overall and strongly relate the treatment schemes (kaletra, DRV + RTV and rezolsta) with the age of the patients as a major criterion in the evolution of the patients. Kaletra and DRV + RTV are inversely correlated with age, while rezolsta is positively correlated with the age of hospitalized COVID-19 patients. Remdesivir was also introduced in the treatment of COVID-19 patients in the second wave according to the adopted protocol, being less correlated with age and negatively connected with gender and the fact that the patient is a smoker or non-smoker and has other comorbidities. Remdesivir is also positively related with kaletra and rezolsta treatment and also with the oxygen saturation. As expected, there is a positive link between the number of hospitalized days and the number of tests until the patient is negative, as well as an inverse correlation between hospitalized days and the evolution of the patient, disease form, symptomatology and oxygen saturation.

The EBICglasso method of estimation allowed us to extract and highlight only the fundamental linkages between considered variables, in a comparative approach between the two waves. Hence, in the first wave, the disease form (DF) was essentially placed in the center of the network between the evolution of the patient (EP) (positive), comorbidities (CMD) (positive), symptomatology (SFA) (positive), DRV + RTV (negative) and rezolsta (positive) treatment, ICU hospitalization (negative), oxygen saturation (SPO2) (negative) and age (positive). In the second wave, the network was configured having the evolution of the patient (EP) in the center and surrounded by disease form (DF) (positive), age (positive), ICU (negative), and oxygen saturation (negative), symptomatology (negative).

These results therefore highlight from an empirical perspective that in the case of the 274 COVID-19 hospitalized patients, the evolution was tightly dependent of the form of the disease, age of the patient, symptomatology, ICU need and oxygen saturation, these coordinates being fundamental in the management of the disease. Treatment was also important, GGMs leading to main correlations merely in the first wave and primarily positive with rezolsta, respectively inversely with DRV+RTV, while kaletra had no connection or one of a much smaller intensity with COVID-19 patient evolution, as empirically attested by GGM results.

### 3.3. Results of Structural Equation Modeling (SEM)

SEM results bring additional empirical evidence to attest that age, gender, disease form and treatment schemes/protocol significantly shape the evolution of the COVID-19 patients. SEM estimations through the maximum likelihood (MLE) procedure for the first and second waves of infection are presented in Figure 4.

By analyzing SEM results we note that in the first wave of infection the treatment schemes tend to have an inverse impact on the evolution of the patient (negative estimated coefficients of −0.034 for DRV-RTV, −0.015 for kaletra, and −0.048 for rezolsta, significant at 1% and 5% thresholds, Figure 4a and Table A2). In the second wave of infection, SEM results entail a positive impact the treatment methods applied (Figure 4b and Table A2, positive estimated coefficients of 0.072 for DRV + RTV, 0.050 for kaletra and 0.103 for rezolsta, respectively Figure 4c and Table A2, positive estimated coefficients of 0.072 for DRV + RTV, 0.046 for kaletra, 0.101 for rezolsta and 0.006 for remdesivir—introduced to hospitalized patients considered in our analysis as a treatment protocol in only in the second wave of infection). At the same time, in the case of both waves (both samples) age and gender positively shape the evolution of hospitalized COVID-19 patients as attested by the positive estimated coefficients (Figure 4a, first wave, 0.027 for age, 0.039 for gender/sex, Figure 4b, second wave, 0.015 for age and 0.14 for gender/sex, Figure 4c, second wave with an additional treatment protocol, 0.015 for age and 0.016 for gender/sex).

The form of disease was also positively associated in a causal relationship with the evolution of the patients in both waves of infection (positive estimated coefficients of 0.041 in the first wave—Figure 4a, 0.059 in the second wave—Figure 4b and 0.060 in the second wave—Figure 4c). The fact that patients needed ICU hospitalization negatively impacted the evolution of the patient in both waves, as expected (negative estimated coefficients of −0.488 in the first wave—Figure 4a, −0.51 in the second wave—Figure 4b,c). We allowed for a correlation between ICU hospitalization, oxygen saturation disease form and the age of the patients in a further impact upon the evolution of the patients and the results brought additional evidence of the interlinkages between these credentials (both positive and negative). All considered variables had a notable impact on the evolution of the COVID-19 hospitalized patients in the case of both waves of infection. These results reinforce previous GGM estimations, as well as the detailed clinical/medical observation and the medical investigations of the patients.

Summarizing, main finding of our research entail that patients in the first wave of infections had a mild form of the disease, with minor symptoms, few of them requiring oxygen therapy. At the same time, patients belonging to the second wave had a much more aggressive form of the disease, with many complications (inaugural diabetes mellitus, hypertension, hematomas located in different areas, pulmonary thromboembolism), many of them had an oxygen saturation at admission into the hospital less than 93% and at discharge a relatively high number of the patients needed O2 concentrator at home. Patients who had severe form of pneumonia remained with pulmonary fibrosis.

## 4. Discussion

In the present single-center study, we analyzed clinical and radiological data, CRP-values, platelet counts and coagulation parameters of 274 COVID-19 patients with pneumonia of different severity in a comparative approach between the two waves of infection focusing on the case of Romania. This study therefore aimed to highlight the differences between the two waves of hospitalizations during the pandemic with COVID-19 in 2020 in Romania. The research endeavor also explores the role of individual features of the patients and various treatment methods in shaping the evolution of hospitalized COVID-19 patients, in a predefined framework.

Following the statistics completed, we found the following differences between the two groups of patients observed.

In the first group the mortality rate was 4.13% compared to the second group where it reached 18.6%. A percentage of 6.89% patients from the first group required invasive mechanical ventilation compared to 10.07% patients from the second batch. The average number of days of hospitalization was 11.87 days in the first wave compared to 12.1 days in the second wave. The average age of people hospitalized in the first batch was 41.86 years, compared to the second one where it was 57.98 years. Additionally, in the first wave 7.58% of patients had a severe form of the disease, while in the second wave 39.53% of patients had a severe form of the disease. Moreover, in the first lot of patients, 4.2% of them had severe pneumonia (CT showed ground glass over 50%), compared to the second lot where the percentage of patients with severe pneumonia was much higher, reaching 28%.

The information can be compared with other recent studies, such as the one conducted between late December 2019, and 26 January 2020 in Wuhan, China by Yang et al. [14]. The data showed 52 critically ill patients included in the study in Wuhan that reached a mortality rate of 61.5% at 28 days of hospitalization.

We found that lymphopenia is common in cases with SARS-COV2 infection. A study conducted by Huang et al. [6] showed that 63% of 41 patients had lymphopenia, while the findings of Yang et al. [14] revealed that lymphopenia occurred in 44 patients (85%). Therefore, individuals who died due to this disease are demonstrated to have had lower lymphocyte level than survivors [15]. Tavakolpour et al. [16] also concluded that lymphopenia occurs more frequently in the severe cases and that there is a higher mortality rate in the elderly. To add to this, Liu et al. [17] highlighted that 22 patients with lymphopenia (40.8%) had a severe or critical form of the disease, and this number was significantly greater than patients with no lymphopenia (9.8%, *p* < 0.001).

Our findings are in line with those of Cappanera et al. [18]. We found that 35.1% of patients from the first group and 92.2% of patients from the second group presented lymphopenia. Zhang et al. [19] entail similar findings, with lymphopenia linked directly with the severity of SARS-COV2.

When it comes to values of D-dimers, 38.6% of patients from the first batch belonging to the first wave had increased values and 64.3% of patients from the second wave showed increased values as well.

Zhang et al. [19] highlight that doctors can predict the mortality level for patients that had D-dimer values of 2.0 µg/mL or higher. Out of the 334 patients that participated in the study, 67 of them had D-dimer ≥ 2.0 µg/mL, and 267 patients with D-dimer < 2.0 µg/mL on admission. 13 patients died, and 12 of them had D-dimer levels ≥ 2.0 µg/mL. This shows a higher incidence of mortality when comparing with those whose D-dimer levels were below 2.0 µg/mL.

Zhou et al. [20] found that out of the 191 cases analyzed, all patients presented increased D-dimer values, but significantly higher values were seen in non-survivors [21].

A meta-analysis performed by Shah et al. [22] that included 18 studies (16 retrospective and 2 prospective) with a total of 3682 patients demonstrated significantly elevated D-dimer levels in patients who died versus those who survived. The risk of mortality was fourfold higher in patients with positive D-dimer versus negative D-dimer (risk ratio, 4.11; 95% CI, 2.48–6.84; *p* < 0.001) and the risk of developing severe disease was twofold higher in patients with positive D-dimer levels versus negative D-dimer (risk ratio, 2.04; 95% CI, 1.34–3.11; *p* < 0.001) [22].

LDH increased levels were seen in the first batch of patients (46.8%) as well as in the second one (82.1%). Chen et al. [23] and Mo et al. [24] presented high values at 20 out of the 29 patients and other 85 severe cases had similar findings.

Another meta-analysis configured by Wibawa-Martha et al. [25] looked at LDH values. From 10,399 patients included in 21 studies, elevated LDH was present in 44% of the patients.

Prothrombin time was increased in both groups of patients covered in this study. 26.8% patients from first wave and 28.6% from the second one showed increased prothrombin time.

Furthermore, on a study conducted on 233 patients, Baranovskii et al. [26] found that prothrombin measured upon admission was prolonged in COVID-19 patients that were further transferred to ICU (82 patients).

The findings of Huang et al. [6] and those of Tang et al. [21] contour own findings grasping that prothrombin time is directly linked to severe cases of SARS-COV2 disease. From 41 cases studied by Huang et al. [6], 13 severe cases in ICU had increased prothrombin time, while Tang et al. [21] presented 183 cases out of which the 21 non-survivors had increased prothrombin time.

Further, 4.13% of patients included in our study showed increased levels of IL-6, 16.5% showed ferritin with increased values, and 37.9% had increased fibrinogen numbers. This finding is similar to the one presented by Zhou et al. [20] that shows elevated levels of serum ferritin and IL-6 in non-survivors, compared with survivors throughout the clinical course.

Remdesivir is an antiviral drug that has been shown to be effective against filoviruses (Ebola) and coronaviruses (SARS-CoV and Middle Eastern respiratory syndrome coronavirus MERS-CoV) which inhibit RNA-dependent RNA polymerase, prematurely blocking RNA transcription [27,28].

In our study, 31% (41 patients) received treatment with remdesivir, out of which 22% died (9 patients), 6 patients needed mechanical ventilation, 5 patients had moderate form of the disease, 36 had severe form of the disease, the mean age of the patients was 57 years. From the total of these patients, 32 of them had an oxygen saturation level below 90% at the check into hospital, and 9 of them up to 93%. The average number of days of hospitalization for patients treated with remdesivir was 14 days until the negative result of RT- PCR COVID-19. Remdesivir was given in combination with corticosteroids (dexamethasone), anticoagulant (fraxiparin), 15 of the patients received tocilizumab and 3 of them received anakinra.

Favipiravir is an antiviral RNA polymerase inhibitor that has been used to treat Ebola and the flu. In our study, 16 patients were treated with favipiravir, 7 of them had a severe form of the disease, and 9 were with a moderate form of the disease, the course of the disease being volatile (no death). The average number of days of hospitalizations was 15 days, until the negative result of the RT-PCR test. The average age of patients was 60.

Favipiravir was given in combination with corticosteroid therapy, anticoagulant, antibiotic therapy and 3 of the patients received anakinra. During COVID-19 infection, a significant number of macrophages and T lymphocytes are activated to produce cytokines, including interleukin-6, responsible for the cytokine storm [29].

Tocilizumab is an anti-human monoclonal antibody, an IL-6 receptor antagonist used in severe forms of COVID-19 infection. In order to receive the drug, patients must meet the following criteria: increased ferritin, decreased lymphocyte and platelet count, increased D-dimers, fibrinogen and CRP [30]. In our study, only 15.5% of patients received treatment with tocilizumab, and they had a severe form of the disease. From this total, 4.6% patients needed mechanical ventilation, the evolution being unfavorable resulting in death. The average age of patients that received 3 doses of tocilizumab was 63. These results are in line with other recent findings of Xu et al. [29] that demonstrate the essential role of tocilizumab in improving the clinical symptoms and repressing the deterioration of severe COVID-19 patients. The drug was administered to 21 patients in severe condition. This was administered among with antiviral therapy (lopinavir/ritonavir/ribavarin); glucocorticoid, other symptom relievers, and oxygen therapy. Another study performed by Salama et al. [31] on 143 patients that received tocilizumab and placebo, revealed that the percentage of patients that received mechanical ventilation or who died was significantly lower in the tocilizumab group than in the placebo group. The same randomized, double blind, placebo-controlled trial involving patients with confirmed severe acute SARS-COV2 infection, tocilizumab was not effective in preventing intubation or death, but patients who received the drug had fewer serious infections than those who received placebo. Most people infected with the new coronavirus tend to have mild to moderate forms of the disease and heal within a few weeks. In contrast, people who survive the COVID-19 infection after needing intubation and longer hospitalization may be left with a number of long-term side effects. COVID-19 infection is proving to be much more than a respiratory illness. It can affect more organs, not just the lungs—from the skin and muscles to the eyes, heart and kidneys—creating long-term health problems. These include fatigue, blurred vision, difficulty breathing, muscle aches, confusion, headaches and even hallucinations.

There is currently a consensus in the medical world that many people who have experienced a severe form of COVID-19 infection require a long recovery period. Some patients discharged develop pulmonary fibrosis and according to the degree of fibrosis, they might require oxygen concentrator at home.

Despite strong data collected, we believe this study has some limitations.

For start, the 274 patients analyzed belong mainly to the West region of the country, more exactly—Timisoara. Patients from other regions were not included in the study. Even though the treatment protocol applied to the 274 patients was in concord with the treatment applied by other hospitals from other parts of the country, we do not have data about the evolution of those patients. Although we believe the sample size was adequate to reach relevant conclusions about the evolution of patients infected with SARS-COV2, we believe that if more research in the literature would have been available, this would have reinforced our findings. Another restriction that we encountered in this study was that the availability of medication was limited. For example, remdesivir was not offered to all patients due to lack of accessibility, therefore, the evolution of disease was different in some patients. The same applies to the drug anakinra. Since the medication was limited at different points in time during the 1st and 2nd waves, the evolution of some patients was different. Another constraint for our results was that treatment protocols have suffered changes. As more and more data became available throughout 2020, and as more knowledge about this virus developed, at national level, different protocols were in place as well as different recommendations. We believe this unpredictability of treatment methods in 2020 is a limitation to our study. Furthermore, considering that we treated the outcome (evolution of the patient—favorable or unfavorable/death) as a binary variable, another limitation of our study is that it may lead to ignore the time-dependent structure of this outcome (time-to-event). This issue is more problematic when traditional methods of logistic and linear regression are employed since these are not suited to be able to include both the event and time aspects as the outcome in the model. As Oh et al. [32] (p. 1276) have also entailed “for time-to-event outcomes, a rich literature exists on the bias introduced by covariate measurement error in regression models”. In this perspective, in order to ensure robust estimates, we designed our research to embed particularly GGMs and structural equation models (SEM) and we also aim to target in our future research SEM with survival-time outcomes, so that exponentiated coefficients can be interpreted as hazard ratios.

## 5. Conclusions

This study explored the evolution of hospitalized COVID-19 patients widely shaped by numerous specific credentials of the disease, personal features of the patients and treatment methods applied, in a comparative approach between the first and second waves of infection during 2020, and by reporting to the particular case of 274 Romanian patients.

Patients in the first wave of infections had a mild form of the disease, with minor symptoms, few of them requiring oxygen therapy.

Looking at most of the biological samples, it can be summarized that patients from both waves had lymphopenia, leucocytosis, thrombocytopenia. However, the percentage of patients showing these findings was lower in the 1st wave, comparing to the 2nd wave, as presented in Table 5. It can be concluded that the 1st wave was milder than the 2nd wave.

Other results in this study back up this conclusion. It can be observed that the increase from 1st wave to 2nd wave is two-fold when it comes to patients presenting elevated levels of fibrinogen and almost five-fold when it comes to elevated results of ferritin. More patients in the 2nd wave presented increased levels of d-dimers, pro-calcitonin, CRP, and IL-6. This finding is supported by results of CT-scans. These investigations showed that in the 1st wave we had around 6% patients with moderate form of pneumonia and a little over 4% of patients with a severe form of pneumonia. A considerably higher number of patients had these complications in the 2nd wave, CT-scans showing 32.55% of patients with moderate form of pneumonia and 30.23% patients with severe form of pneumonia. Moreover, to support the fact that the 2nd wave was much more aggressive than the 1st wave, almost 40% of patients observed in the 2nd wave had a severe form of the disease, compared to 7.58% patients from the 1st wave. The same result applies to the sample of critically ill patients, with 17.30% in the 2nd wave compared to only 3.44% in the 1st wave. Additionally, a higher percentage of patients needed intensive care unit, and this was followed by a higher number of deaths (with 4.13% of patients losing their life in the 1st wave compared to 18.6% in the 2nd wave).

Patients from the 2nd wave had a much more aggressive form of the disease, with many complications (inaugural diabetes mellitus, hypertension, hematomas located in different areas, pulmonary thromboembolism), many of them had an oxygen saturation at admission into the hospital less than 93% and at discharge a relatively high number of the patients needed O2 concentrator at home. Patients who had severe form of pneumonia remained with pulmonary fibrosis.

The damage in the infection with COVID-19 is multiorganic. Although at first it was thought to affect only the respiratory system, with time it was discovered that the liver, pancreas, kidneys, heart and central nervous systems are affected as well. During hospitalization there were patients who had depression, memory loss, disorientation without a documented neurological impairment.

It is for this reason we believe that the treatment of patients with COVID-19 infection is multidisciplinary, requiring, participation of doctors with specializations in infections, pulmonologists, cardiologists, diabetologists, neurologists, surgeons and nephrologists.

The research therefore brings new evidence to strengthen the knowledge in this field and presents a comprehensive two-fold assessment (medical observation/investigation and econometric modeling through GGMs and SEM) of the evolution of COVID-19 patients in a particular setting, predefined framework, as largely detailed within the paper. Both advanced methods to modeling longitudinal data, GGMs and SEM, have provided important insights on the specific ways in which the individual features of the patients and the specific treatment methods applied can positively influence the evolution of COVID-19 patients. These complex and modern research methods are complementary, hence combined in such a way as to enhance the qualities of each other, each trying to discard the other’s limits, so that the final estimations are accurate, robust, and correctly interpreted and support the conclusions drawn. In case of the first wave of COVID-19 infection, GGM results entail that there is a strong positive correlation between the evolution of the patients and the COVID-19 disease form, which is further positively correlated with the treatment scheme. The evolution of the patients is strongly and inversely correlated with the symptomatology and the ICU hospitalization. Moreover, the disease form is strongly and inversely correlated with oxygen saturation and the residence of patients (urban/rural). Age and gender are also important credentials that shape the disease form and patient evolution in responding to treatment as well.

The size of the sample may represent a limitation of the current research, yet the study aims to provide new guidelines in the field and strengthens current knowledge; hence, hard work was invested to accurately observe and treat these patients, as well as to perform the study. Future research targets a detailed analysis on sub-samples according to age and gender of the patients as to expand the assessment on patients’ evolution under various treatment protocols enforced, with a separate research design comprising mixed graphical models (MGMs) and SEM with survival-time outcomes as main methodological credentials.

## Figures and Tables

**Figure 1 jcm-10-01958-f001:**
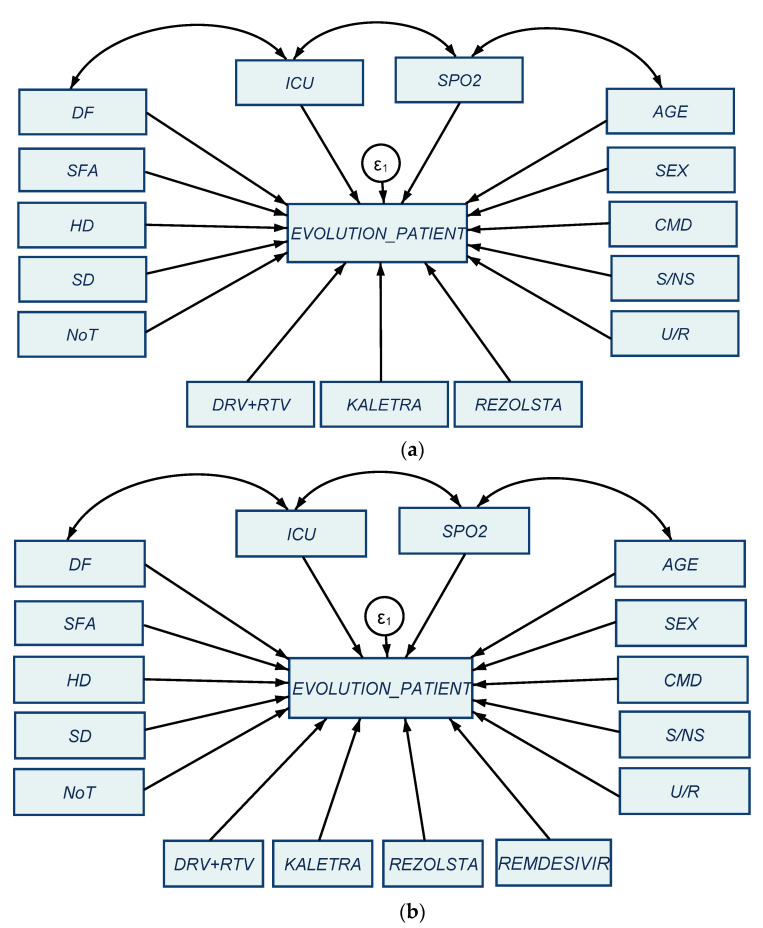
General configuration of the SEM models: (**a**) first sample of 145 COVID-19 patients from the first wave; (**b**) second sample of 129 COVID-19 patients from the second wave. Source: Own configuration in Stata 16.

**Figure 2 jcm-10-01958-f002:**
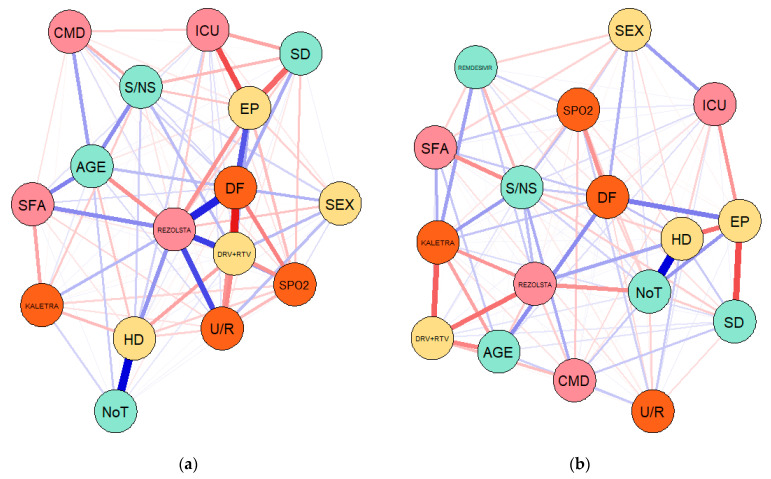
GGMs configuration through partial correlations method and the associated results: (**a**) first sample of 145 COVID-19 patients from the first wave of infection; (**b**) second sample of 129 COVID-19 patients from the second wave of infection. Source: Own configuration in RStudio version 3.6.3.

**Figure 3 jcm-10-01958-f003:**
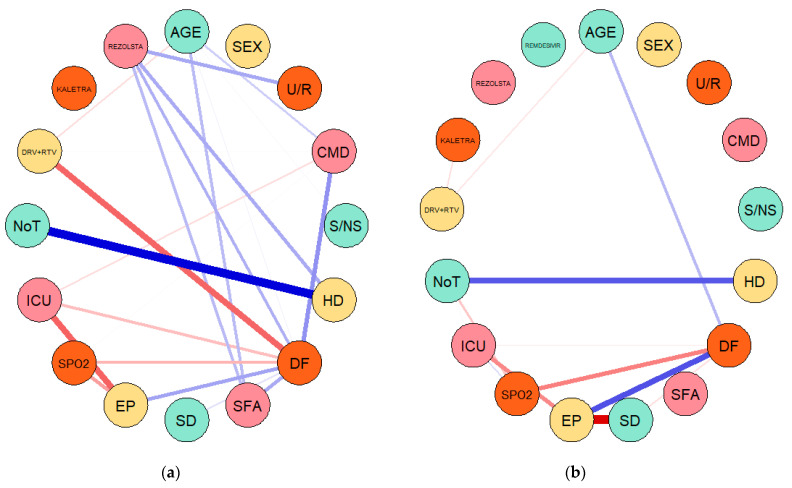
GGMs configuration through the EBIC with graphical lasso method and the associated results: (**a**) first sample of 145 COVID-19 patients from the first wave of infection; (**b**) second sample of 129 COVID-19 patients from the second wave of infection. Source: Own configuration in RStudio version 3.6.3.

**Figure 4 jcm-10-01958-f004:**
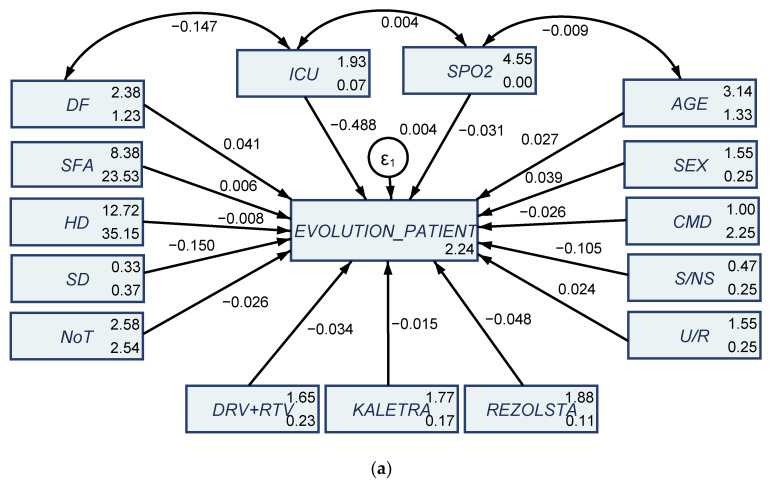
SEM results obtained through the maximum likelihood estimation method: (**a**) first sample of 145 COVID-19 patients from the first wave of infection; (**b**) second sample of 129 COVID-19 patients from the second wave—first treatment scheme; (**c**) second sample of 129 COVID-19 patients from the second wave—second treatment scheme with Remdesivir. Source: Own configuration in RStudio version 3.6.3.

**Table 1 jcm-10-01958-t001:** Description of individual features, demographics and other characteristics of patients analyzed in both samples (a comparative approach between the two waves of infection with SARS-COV2 in Romania during 2020).

	No. Patients	Gender	Age (Average)	Residence (Urban/Rural)	Smokers/Non-Smokers	Days of Hospitalization (Average)	Disease Form	Deaths/ICU
1st wave	145	76 men (52%); 69 women (48%)	42 years	55.6% urban; 44.4% rural	48.9% smokers; 51.1% non-smokers	12 days	53.1%—mild; 25.51%—moderate; 7.58%—severe; 3.44%—critical; 10.37%—asymptomatic	6 patients died (4.13%); 10 patients were treated in the ICU (6.89% of patients)
2nd wave	129	61 men (47%); 68 women (53%)	58 years	58.9% urban; 41.1% rural	30.2% smokers; 69.7% non-smokers	12 days	6.97%—mild; 35.65%—moderate; 39.53%—severe; 17.08%—critical; 0.77%; asymptomatic	24 patients died (18.60%); 13 patients were treated in the ICU (10.07% of patients)

Source: Authors’ contribution.

**Table 2 jcm-10-01958-t002:** First sample of COVID-19 patients from the first wave of infection.

	N	Mean	Var	Sd	Min	Max
AGE	145	42	348.0	18.6	1	83
HD	145	12	31.8	5.6	2	29
SPO2	139	95	21.7	4.6	70	112
NoT	143	2	3.0	1.7	1	9
N total	145					

Source: Authors’ contribution.

**Table 3 jcm-10-01958-t003:** Second sample of COVID-19 patients from the second wave of infection.

	N	Mean	Var	Sd	Min	Max
AGE	129	58	310.8	17.6	1	91
HD	129	12	53.3	7.3	2	57
SPO2	129	89	52.5	7.2	50	99
NoT	102	2	0.9	0.9	1	6
N total	129					

Source: Authors’ contribution.

**Table 4 jcm-10-01958-t004:** Symptomatology and comorbidities of 274 SARS-COV2 patients considered in the analysis.

Symptomatology at Onset (SFA)	1st Wave	2nd Wave
Fever	62.06%	66.66%
Headache	37.93%	33.33%
Cough	66.89%	79.06%
Anosmia and ageusia	13.1%	14.72%
Myalgias	27.58%	27.13%
Dysphonia	4.13 %	9.3%
Dysphagia	13.7%	8.52%
Shivers	12.4%	13.17%
Digestive symptoms	16.55%	16.27%
Dyspnoea	15.17%	51.16%
**Symptomatology during the hospitalization**	**1st wave**	**2nd wave**
Cough	82.75%	62.56%
Fever	11.03%	14.72%
Dyspnoea	20%	72.86%
Headache	5.51%	10.1%
Anosmia and ageusia	12.4%	7.75%
Myalgia	15.17%	5.4%
Dysphagia	16.55%	6.8%
Digestive symptoms	7.58%	4.65%
Asthenia (fatigue)	16%	62.79%
**Symptomatology at discharge (SD)**	**1st wave**	**2nd wave**
Cough	20%	2.3%
Oxygen concentrator at home	5.5%	26.13%
Asthenia (fatigue)	8%	18.6%
**Comorbidities at onset (CMD)**	**1st wave**	**2nd wave**
Hypertension	26.89%	70.54%
Hashimoto‘s thyroiditis	4.82%	5.42%
Type 2 diabetes	10.34%	27.13%
Heart failure	3.44%	37.20%
Obesity	6.89%	23.25%
Sepsis	4.13%	19.37%
Chronic kidney disease	4.13%	9.3%
Cerebrovascular accident	2%	4.6%
Atrial fibrillation	1.37%	5.4%
Asthma	1.37%	5.4%
Acute respiratory failure	20%	53.48%
Mixed dementia	0%	6.9%
**Comorbidities developed during hospitalization**	**1st wave**	**2nd wave**
Sinus tachycardia	0%	1.55%
Hepatic cytolysis syndrome	4.82%	34.10%
Inaugural diabetes	6.2%	38.75%
Acute renal failure	6.89%	11.62%
Pleurisy	1.37%	0%
Pulmonary emphysema	1.37%	0%
Pulmonary thromboembolism	0%	2.3%
Clostridium difficile enterocolitis	0%	2.3%
Upper digestive hemorrhage	0%	0.77%

Source: Authors’ contribution.

**Table 5 jcm-10-01958-t005:** Biological samples and treatment schemes for the 274 SARS-COV2 patients considered in the analysis.

Biological Samples and Treatment from 274 Hospitalized Patients	1st Wave	2nd Wave
Leukopenia	15.1%	18.6%
Leucocytosis	11.7%	43.4%
Thrombocytopenia	15.17%	17.0%
Thrombocytosis	24.8%	12.4%
Lymphopenia	35.1%	92.2%
Hemoglobin ↓	13.1%	25.5%
Prothrombin time ↑	26. 8%	28.6%
Fibrinogen ↑	37.9%	82.1%
D-Dimers ↑	38.6%	64.3%
Ferritin ↑	16.5%	81.3%
LDH ↑	46.8%	82.1%
Urea↑	15.1%	29.4%
Creatinine ↑	21.3%	14.72%
Hyponatremia	46.2%	46.5%
Hypokalemia	4.13%	4.6%
Hyperkalemia	15.8%	6.75%
Aspartate aminotransferase (SGOT/AST) ↑	44.8%	38.7%
Alanine aminotransferase (GPT/ALT) ↑	48.2%	45.7%
Procalcitonin ↑	4.13%	19.37%
Hyperglycemia	34.4%	82.1%
Lactate↑	19.3%	42.6%
CRP↑	47.5%	89.9%
Interleukin 6↑	4.13%	75.1%
Troponin↑	2%	2%
CT scan	16%—mild form of pneumonia 6.4%—a moderate form of pneumonia 4.2%—a severe form of pneumonia	25.58%—mild form of pneumonia 32.55%—moderate form of pneumonia 30.23%—severe form of pneumonia
Antiviral medication	Lopinavir/Ritonavir—25.65%, Darunavir + Cobicistat—24.58%, Darunavir + Ritonavir—41.13%, Hydroxychloroquine—28.96%	Lopinavir/Ritonavir—3.87%, Darunavir + Cobicistat—4.65%, Darunavir + Ritonavir—79.06%, Favipiravir—12.50%, Remdesivir—31%
Antibiotic therapy	Azithromycin—35.17% Ceftriaxone—21.37% Vancomycin—2.78%	Ceftriaxone—46.51%, Ceftriaxone + Levofloxacin—3.87%, Ceftriaxone + Moxifloxacin—4.65%, Vancomycin—1.55%, Meropenem + Vancomycin—10.07%
Corticotherapy	31%	77.51%
Anticoagulant	21.37%	94.57%
Immunomodulatory medication	0%	Anakinra—4.65% Tocilizumab—15.50%

Source: Authors’ contribution.

## Data Availability

The dataset can be provided by the authors upon request.

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
