# Peer review of "Individual Characteristics as Prognostic Factors of the Evolution of Hospitalized COVID-19 Romanian Patients: A Comparative Observational Study between the First and Second Waves Based on Gaussian Graphical Models and Structural Equation Modeling"

_jcm, 2021, doi:10.3390/jcm10091958_

Round 1
Reviewer 1 Report
The authors have reviewed the suggestions made by the reviewers.
The article has now been improved from the previous one.
n.
Author Response
Thank you very much for your encouraging feedback!
We are very grateful for investing your time to analyze the paper and for your accurate and thoughtful comments that were essential in enhancing the overall research and the final version of the manuscript.
Reviewer 2 Report
Although the paper has been improved by the revision, I still have serious concerns from a statistical perspective. As already mentioned in my last review, GGMs are designed for normally distributed variables and should not be used for e.g. count or binary data as is done in the present manuscript. More robust alternatives are Mixed Graphical Models, which are also discussed in the paper by Altenbuchinger et al. which the authors cite.
Apart from this, I have some minor comments:
1. I still don’t see the merit of Figure 1 and suggest the authors remove it in order to prevent confusion.
2. Disease severity contains a category “asymptomatic patients”. However, all data was collected within hospitals. Were these COVID-19 patients detected by chance alone or how are they classified as “asymptomatic”?
3. Table 1: This table has greatly improved. However, I would suggest to round the results to 1 digit in order to increase readability. Moreover, sensible decimals should be specified: e.g. 11.87 days corresponds to 11 days, 20 hours, 52 minutes and 48 seconds… I don’t assume measurements were taken on a level of seconds? Also, please remove redundant information such as the number of women if the number of men is given.
4. Table 1: For the first wave, to numbers for residence don’t sum to 100%. Why is that?
5. Table 2 and 3: what does e(Sum) refer to? Please explain or remove this column.
6. Figure 3 and 4: it does not become clear what the different subplots refer to. Why are there two plots for each wave and each method?
7. The authors present different models with partly differing conclusions. Some discussion of which model to believe or how they differ would be appropriate.
8. The definition of “evolution of a patient” is still not entirely clear. Does this refer to death yes/no? Because ICU is considered as a separate variable. Since this variable is the outcome, it should be specified more precisely.
Author Response
Dear Reviewer,
Thank you very much for the opportunity to reconsider the manuscript and to undertake major revisions, marked with track changes into the document, by addressing the observations received! We performed several improvements that have added significant value to our research endeavour. We are very grateful for investing your time to analyse the paper and make very accurate, encouraging and thoughtful comments and recommendations. Thus, in accordance with the observations received, we performed the following revisions:
- Regarding the observation that „Although the paper has been improved by the revision, I still have serious concerns from a statistical perspective. As already mentioned in my last review, GGMs are designed for normally distributed variables and should not be used for e.g. count or binary data as is done in the present manuscript. More robust alternatives are Mixed Graphical Models, which are also discussed in the paper by Altenbuchinger et al. which the authors cite” – thank you very much! The authors would like to entail that the research endeavor performed in this manuscript relies on a two-fold approach, namely a detailed clinical observation and investigation of patients with Covid-19, along with a quantitative assessment through network analysis performed with the use of GGMs processed through 2 estimation methods EBICglasso and Pcor in order to ensure robustness of the results and structural equation modelling – SEM models processed through the maximum likelihood estimator (MLE). These complex and modern research methods are complementary, hence combined in such a way as to enhance the qualities of each other and therefore enforce the results obtained and support the conclusions drawn. For this particular reason, in this empirical endeavor, we applied numerous econometric techniques, each trying to discard the other’s limits, so that the final estimations are accurate, robust, and correctly interpreted. We have selected these research methods from a variety of methods based on the theoretical and empirical groundings supporting this research, including previous articles published by the authors. Our empirical analysis therefore relied on two approaches to modelling longitudinal data, namely Gaussian graphical models (GGMs) and structural equation modelling (SEM). GGMs are among the most popular methods to infer networks from various data and they became standard analysis tools as also stated by Altenbuchinger et al. (2020). Both techniques GGMs and SEM imply a variance-covariance matrix, aiming to identify how variables are related to each other, namely the direct and indirect effects of one variable on another, having their origin in path analysis. The main advantage of the GGMs is the ability to handle different types of variables, as comprised in our dataset, since the variables included in the empirical analysis had different measurement units (e.g. binary, multi-category), while the SEM brings forward our research since it rejoins path analysis, factor analysis and regression, thus allowing to specify multiple causal associations between our constructs. Hence, we were able to model conditional associations, namely the degree in which the variables are independent after conditioning on all other variables in the data set (regardless of the fact that these variables have different measurement units). All of these credentials need to be assessed in their tight interdependence and sequential approach, as a complex network (performed in this paper through the GGMs) and through causal relationships (as enhanced by the SEM models designed to achieve the complicated model setup).
On these lines and considering the observation received, we have processed the graphical models on both samples of Covid-19 patients with the use of the RStudio-package and through the mgm command specific for Mixed Graphical Models (MGMs), in order to estimate the conditional independence network in our datasets. The MGMs were also processed through EBICglasso and Pcor and the results obtained are similar with those of the GGMs as reported in our manuscript.
We entail these MGMs here (please see the attachment), since we did not capture them within the manuscript because there are no differences between the two sets of graphical models, GGMs and MGMs.
Furthermore, we have stated in the Conclusion section, as future research directions, that we will focus on a separate research design comprising Mixed Graphical Models (MGMs) as main methodological credentials.
- Regarding the observation that: „1. I still don’t see the merit of Figure 1 and suggest the authors remove it in order to prevent confusion” – thank you very much! We have deleted Figure 1 from the manuscript.
- As regards the observation that „2. Disease severity contains a category “asymptomatic patients”. However, all data was collected within hospitals. Were these COVID-19 patients detected by chance alone or how are they classified as “asymptomatic”?” – In the time frame included in this study, many hospitals in Romania received symptomatic and asymptomatic patients. The asymptomatic patients did not present any symptoms and they were only admitted based on a positive PCR test. During those months, hospitals in Romania (including Infectious Diseases and Pneumo-phthisiology Hospital Timisoara) served also as isolation units and this is the reason why asymptomatic patients were checked in. Thank you very much!
- Regarding the observation that „3. Table 1: This table has greatly improved. However, I would suggest to round the results to 1 digit in order to increase readability. Moreover, sensible decimals should be specified: e.g. 11.87 days corresponds to 11 days, 20 hours, 52 minutes and 48 seconds… I don’t assume measurements were taken on a level of seconds? Also, please remove redundant information such as the number of women if the number of men is given” – we performed all these corrections as indicated. Thank you very much!
- As regards the observation that „4. Table 1: For the first wave, to numbers for residence don’t sum to 100%. Why is that?” – thank you very much for highlighting this to us, we corrected this error.
- Regarding that „5. Table 2 and 3: what does e(Sum) refer to? Please explain or remove this column” - we removed this column, thank you very much!
- As regards „6. Figure 3 and 4: it does not become clear what the different subplots refer to. Why are there two plots for each wave and each method?” – thank you very much for entailing this to us! We deleted the redundant subplots and reported only one plot per each wave and each method.
- As regards the observation that „7. The authors present different models with partly differing conclusions. Some discussion of which model to believe or how they differ would be appropriate” – thank you very much! We added several lines on this topic in the Conclusion section.
- Regarding that „8. The definition of “evolution of a patient” is still not entirely clear. Does this refer to death yes/no? Because ICU is considered as a separate variable. Since this variable is the outcome, it should be specified more precisely” – Yes, the unfavorable evolution of the patient refers to death. Not all the patients that developed serious complications and were brought in the ICU department have died and for this reason EP (evolution of the patient) is a separable variable than ICU. We tried to state it clearer within the manuscript and hope this clarified the confusion. Thank you very much!

Round 2
Reviewer 2 Report
Thank you for the clarifying comments and the changes applied to the manuscript. I only have one minor final comment: By treating the outcome (death) as a binary variable, you ignore the time-dependent structure of this outcome (time-to-event). It would be good to include a short sentence in the discussion mentioning this issue.
Author Response
Thank you very much for your encouraging feedback!
We included a short sentence in the discussion section mentioning the issue of time-dependent structure of the outcome (time-to-event).
This manuscript is a resubmission of an earlier submission. The following is a list of the peer review reports and author responses from that submission.
Round 1
Reviewer 1 Report
- The title should contain information about the method used in the article.
- - The abstract should contain more data on the results. The main part is the introduction and the methods in the introduction.
- the introduction is long and should better justify the manuscript.
- The objectives of the work should be written in the introduction section.
- The table of materials and methods should be included in the results.
- It should include part of the text in a table.
- Discussion, it should include the limitation of the study
- The conclusion should be more concrete
Reviewer 2 Report
This study analyses potential factors influencing disease progression/severity in COVID-19
patients. However, the methods applied are inappropriate for the data at hand and the presentation
of the results is very confusing. See my specific comments below.
Comments:
1. Table 1: The presentation of Table 1 is a catastrophe. Continuous and categorical variables
are mixed and means and variances are presented for categorical variables (e.g. sex, U/R, S/
NS, …). This type of presentation is completely inappropriate and not at all informative. For
example, how should a mean comorbidity of 0.95 be interpreted? Let alone a mean sex of
1.52… Categorical variables must be stated as n (%) or median (IQR) were appropriate.
2. Figure 1: What is the interpretation of this plot? What do the numbers on the outside of the
circle refer to? How should the two waves be compared?
3. Methodology: You applied Gaussian Graphical Models to detect associations in the data. As
stated in the methodology section on page 5, however, these are designed for continuous and
normally distributed random variables. However, you apply them to all types of data
including nominal variables such as sex.
4. The description of the GGM remains unclear. Some formulas are presented but the
connection to the model itself remains unclear and the explanations sparse.
5. Page 7, line 234: what were these patients randomized to? Different treatments? If yes,
which ones?
6. The results section 3.1 is impossible to read. A lot of numbers and percentages are reported
(some being redundant, e.g. it suffices to report the percentage of men which allows to
calculate the percentage of women in the study). These results should better be summarized
in a table, which would also allow to view the two waves side-by-side.
7. It does not make sense to consider all these variables in the analysis without further
specification. For example, it is not surprising that the number of tests and the number of
hospitalized days are strongly correlated. Also, how does disease form and patient evolution
differ? Which comorbidities were considered? How are the variables SFA and SD
calculated? This needs to be explained in more detail.
8. Figure 5: It remains unclear, what the numbers reported refer to.
9. Discussion: This seems more like a part of the Results section than a discussion. Also
references for some statements are missing, e.g. studies showing the long-term health
problems of COVID-19 patients.
10. Table A2: The merit of this table is unclear to me. It is almost impossible to read and some
of the abbreviations are not defined or unclear (e.g. what does the last column refer to? What
is ati? Or _cons?)
11. The variable names are not consistent. Evolution of the patient is sometimes abbreviated EP
and sometimes PE. Also, it should be noted at the beginning where the abbreviations can be
found.